# Genome-Wide DNA Methylation Profiling as a Prognostic Marker in Pituitary Adenomas—A Pilot Study

**DOI:** 10.3390/cancers16122210

**Published:** 2024-06-13

**Authors:** Morten Winkler Møller, Marianne Skovsager Andersen, Bo Halle, Christian Bonde Pedersen, Henning Bünsow Boldt, Qihua Tan, Philipp Sebastian Jurmeister, Grayson A. Herrgott, Ana Valeria Castro, Jeanette K. Petersen, Frantz Rom Poulsen

**Affiliations:** 1Department of Neurosurgery, Odense University Hospital, 5000 Odense, Denmark; bo.halle@rsyd.dk (B.H.); christian.bonde@rsyd.dk (C.B.P.); frantz.r.poulsen@rsyd.dk (F.R.P.); 2Department of Clinical Research and BRIDGE (Brain Research—Inter Disciplinary Guided Excellence), University of Southern Denmark, 5000 Odense, Denmark; marianne.andersen1@rsyd.dk (M.S.A.); henning.boldt@rsyd.dk (H.B.B.); qtan@health.sdu.dk (Q.T.); jeanette.krogh.petersen@rsyd.dk (J.K.P.); 3Department of Endocrinology, Odense University Hospital, 5000 Odense, Denmark; 4Department of Pathology, Odense University Hospital, 5000 Odense, Denmark; 5Department of Public Health, Odense University Hospital, 5000 Odense, Denmark; 6Institute of Pathology, Ludwig Maximilians University Hospital Munich, 80336 Munich, Germany; philipp.jurmeister@med.uni-muenchen.de; 7German Cancer Consortium (DKTK), Partner Site Munich, and German Cancer Research Center (DKFZ), 69120 Heidelberg, Germany; 8Omics Laboratory, Hermelin Brain Tumor Center, Department of Neurosurgery, Henry Ford Health, Detroit, MI 48202, USA; gherrgo1@hfhs.org (G.A.H.); acastro1@hfhs.org (A.V.C.); 9Department of Physiology, College of Human Medicine, Michigan State University, E. Lansing, MI 48824, USA

**Keywords:** pituitary adenoma, nonfunctioning pituitary adenoma, DNA methylation, regrowth potential, recurrence, epigenetics

## Abstract

**Simple Summary:**

Pituitary adenomas often regrow after surgery, posing a treatment challenge. By studying the DNA methylation patterns of these adenomas, we aimed to identify different subtypes that may predict regrowth potential. Samples from 33 patients with nonfunctioning pituitary adenomas were analyzed, revealing distinct DNA methylation clusters. While trends suggesting certain clusters might be associated with higher regrowth potential emerged, overall significant differences in regrowth rates between these clusters were not found. DNA methylation profiling could help identify adenomas prone to regrowth, potentially improving treatment strategies in the future. Further larger studies are needed to confirm the findings from this explorative pilot study.

**Abstract:**

Background: The prediction of the regrowth potential of pituitary adenomas after surgery is challenging. The genome-wide DNA methylation profiling of pituitary adenomas may separate adenomas into distinct methylation classes corresponding to histology-based subtypes. Specific genes and differentially methylated probes involving regrowth have been proposed, but no study has linked this epigenetic variance with regrowth potential and the clinical heterogeneity of nonfunctioning pituitary adenomas. This study aimed to investigate whether DNA methylation profiling can be useful as a clinical prognostic marker. Methods: A DNA methylation analysis by Illumina’s MethylationEPIC array was performed on 54 pituitary macroadenomas from patients who underwent transsphenoidal surgery during 2007–2017. Twelve patients were excluded due to an incomplete postoperative follow-up, degenerated biobank-stored tissue, or low DNA methylation quality. For the quantitative measurement of the tumor regrowth rate, we conducted a 3D volumetric analysis of tumor remnant volume via annual magnetic resonance imaging. A linear mixed effects model was used to examine whether different DNA methylation clusters had different regrowth patterns. Results: The DNA methylation profiling of 42 tissue samples showed robust DNA methylation clusters, comparable with previous findings. The subgroup of 33 nonfunctioning pituitary adenomas of an SF1-lineage showed five subclusters with an approximately unbiased score of 86%. There were no overall statistically significant differences when comparing hazard ratios for regrowth of 100%, 50%, or 0%. Despite this, plots of correlated survival estimates suggested higher regrowth rates for some clusters. The mixed effects model of accumulated regrowth similarly showed tendencies toward an association between specific DNA methylation clusters and regrowth potential. Conclusion: The DNA methylation profiling of nonfunctioning pituitary adenomas may potentially identify adenomas with increased growth and recurrence potential. Larger validation studies are needed to confirm the findings from this explorative pilot study.

## 1. Introduction

Pituitary adenomas (PAs) arise from the anterior lobe of the pituitary gland [1]. The updated WHO classification from 2017 provided a more precise pathological classification of PAs by including immunohistochemical staining based on the transcription factors SF1, PIT1, and TPIT, which regulate the differentiation of adenohypophysial cells [2]. This transcription factor staining correlates with a methylome analysis that divided PAs into three distinct subgroups. TPIT (TBX19) is expressed in corticotrophs (ACTH-producing and rarely clinically silent adenomas), PIT1 (POUF1) is expressed in GH-, PRL-, and TSH-producing adenomas (rarely clinically silent), and SF1 (NR5A1) is expressed in gonadotropinomas, the majority clinically nonsecreting [3,4]. The term nonfunctioning pituitary adenomas (NFPAs) covers all clinically silent adenomas. Regrowth potential has been corelated with transcription factor staining, where PIT1- and TPIT-lineage NFPAs have been associated with a higher regrowth rate and more invasive behavior [5]. Transcription factor staining has improved the overall prognostication and treatment of PAs [6]; however, diagnosis and prognostication are still a challenge for NFPAs. 

DNA methylation is a continuous process that regulates gene expression at the epigenetic level. This regulation is facilitated by enzymes called DNA methyltransferases that transfer a methyl group to the C5 position of cytosine nucleotides in DNA, resulting in the formation of 5-methylcytosine. When 5-methylcytosine occurs before a guanine nucleotide in DNA sequences, these regions are referred to as CpG sites. The accumulation of CpG sites forms what are known as CpG islands. Approximately 70% of gene promoters are found within CpG islands [7], which are evolutionarily conserved and play a crucial role in gene regulation, particularly concerning chromatin structures [7]. The methylation of CpG islands has led to the silencing of gene expression, resulting in tumor regrowth through the silencing of inhibitors [8]. The segregation of different tumors based on DNA methylation is based on differentially methylated positions (DMPs) [9], with the accumulation of DMPs in a genomic region defined as differentially methylated regions (DMRs) [10]. These differential patterns enable the generation of a distinct methylation profile for each individual tumor, making each sample’s profile remarkably unique. Despite this individuality, similarities among tumors within certain subgroups have been proven to be valuable for tumor classification, enabling differentiation between malignant and benign tumors [11].

Using genome-wide DNA methylation profiling, PAs have been separated into distinct methylation classes, typically corresponding to histology-based subtypes (PIT1-, TPIT- and SF1-lineages) [4,12,13,14]; however, further evidence of the clinical significance of these methylation classes in PAs is currently lacking, and the predictive value of the molecular features in terms of clinical outcomes is largely unknown. Previous studies [15,16,17,18,19,20,21] have uncovered clinically significant genes and related DMPs that are potentially implicated in tumor regrowth, but to our knowledge no studies have correlated specific subtypes of NFPAs with regrowth potential. Multiple studies indicate that invasive tumors and postoperative residual tumor tissue increases the risk of regrowth [22]. Several studies show the regrowth of PAs, in which the definition is the enlargement of residual tumors in those with subtotal resection or recurrence in those with gross total resection [23]. There are currently no prognostic markers available with which to predict these recurrences or whether a subsequent reintervention will be needed.

This retrospective study aimed to cluster SF1-lineage NFPAs, described simply as NFPAs in this paper, according to DNA methylation status and to correlate these data with regrowth rates, immunohistochemical staining, functional status, and reintervention (reoperation or radiotherapy), to investigate whether DNA methylation profiling can be useful as a clinical prognostic marker.

## 2. Materials and Methods

### 2.1. Patient Inclusion

Patients operated on for PAs at the Department of Neurosurgery, Odense University Hospital, from 2007 to 2017 were included in the study. All patients operated on for PAs are followed in a joint endocrinological and neurosurgical outpatient clinic, with annual follow-ups of clinical, radiological, and biochemical status. The study inclusion criteria were a complete medical history before surgery, a clinical presentation of NFPAs (corresponding to a histological assessment based on the 2004 classification and negative for hormone secretion), and macroadenomas with tumor volumes of 3–7 cm^3^ to ensure sufficient tissue for a DNA methylation analysis. Included patients operated on prior to 2007 for PAs but with another surgical procedure in the study period were considered secondary surgeries.

### 2.2. Radiological Classification

Annual magnetic resonance imaging (MRI) allowed for the 3D volumetric analysis of possible tumor remnants using the program Horos® (v4.0.1), as previously shown [24,25]. Tumor volume was calculated using a 3D volumetric analysis based on axial, frontal, or sagittal sections on T1-weighted images, preferably with contrast. Similarly, tumor remnant volume was calculated for all follow-up MRIs. A quantitative measure of the tumor regrowth rate was determined by the change in the tumor remnant volume as the percentage accumulated over time.

The tumors were preoperatively classified based on the Hardy and Knosp classification systems [26,27,28]. Invasive growth was characterized by Hardy’s classification of 3 or 4 and/or D or E, as well as by Knosp’s classification of 3 or 4. 

### 2.3. DNA Methylation

For each patient, we used archived formalin-fixed and paraffin-embedded (FFPE) tissue samples stored at the Department of Pathology, Odense University Hospital. HE-stained sections were reviewed for representative adenoma tissue. DNA was isolated and purified from the FFPE tissue blocks using a Qiagen Generead DNA FFPE Kit (QIAGEN, Hilden, Germany), and a minimum of ~50 ng of DNA underwent bisulfite conversion and methylation analysis on Illumina’s MethylationEPIC array (Illumina Inc., San Diego, CA, USA) according to the manufacturer’s protocol. See the Appendix A for further details.

Based on the raw Intensity Data files (IDAT-files), the DNA methylation analysis was performed using the *minfi* (v1.48.0) and *watermelon* (v2.8.0) packages in R (v3.17) for data preprocessing with beta values and M-values. A quality control report was generated to assess data quality, and the rgSet.filt function was used to discard probes and samples with poor quality. Normalization was performed by the *preprocessQuantile* function, and additional preprocessing was performed using the *dasen* function. X/Y chromosome probes, SNPs, and cross-reactive probes were removed. An unsupervised hierarchical cluster analysis (UHCA) was performed using the *hclust* function in R. Additional clustering was carried out on SF1-lineage nonfunctioning pituitary adenomas only via a bootstrap analysis (×200) on the preprocessed beta values using the *pvclust* package (v2.2-0). Approximately unbiased (AU) and bootstrap probability (BP) values are measures of support for clusters in hierarchical clustering, and these values range from 0 to 1. High AU and BP values indicate strong support for the corresponding clusters. A heatmap was generated using the *ComplexHeatmap* package (v2.18.0) in R for the 5000 most variable CpG sites based on beta values. Differentially methylated positions (DMPs) were identified comparing samples from one cluster to the remaining samples in the cohort of SF1-lineage NFPAs, using significant adjusted *p*-values (by the false discovery rate, FDR) and the mean methylation difference across pairwise comparisons (Wilcoxon rank-sum test). Additionally, cutoffs at 0.05 for adjusted *p*-values and > 0.2 for methylation differences were set for significant DMPs. Differentially methylated regions (DMRs) were calculated using the *DMRcate* package (v2.16.1), with a cutoff above 0.30 in the mean difference in beta values for identifying significant DMRs.

The functional enrichment analysis of DMPs and DMRs in relation to potential gene overlaps was carried out in the Molecular Signatures Database (MsigDB) from Gene Set Enrichment Analysis (GSEA), developed by the Broad Institute [29], and the Genomic Regions Enrichment of Annotations Tool (GREAT), developed at the Stanford School of Medicine by the Bejerano Lab [30]. DMPs from each methylation cluster were imported into GSEA and cross-referenced with gene overlaps for each DMP.

Similarly, all DMRs for each cluster were imported into the GREAT and were assigned linkage if the regions were located within 1000 kb upstream or downstream of a gene’s DNA sequence.

### 2.4. Statistical Analysis

An ANOVA was performed on age at surgery, preoperative tumor volume, postoperative tumor volume, and median follow-up in months between methylation clusters.

Kaplan–Meier survival estimates were computed to test whether different DNA methylation clusters possessed unique growth rates. Four plots were generated based on four different events: (1) reintervention, (2) no regrowth, (3) low regrowth rate, and (4) high regrowth rate. Reintervention was defined as either reoperation or radiotherapy in the study period. No regrowth was defined as no growth during the study period, a low regrowth rate was defined as less than 50% tumor volume expansion, a medium regrowth rate was defined as 51–100% tumor volume expansion, and a high regrowth rate was defined as more than 100% tumor volume expansion during the study period. Hazard ratios were calculated through a Cox proportional hazard model. 

A linear mixed effects model was employed to examine whether different DNA methylation clusters had different regrowth patterns. The analysis accounted for preoperative tumor volume, age at first surgery, reoperation, and additional radiotherapy. The model included adjustments for reintervention, preoperative tumor volume, and age at surgery, with a random intercept for each patient. The analysis was limited to up to eight years postoperatively, as many patients had irregular follow-ups beyond eight years. A 95% confidence interval (95% CI) was computed for each difference between the DNA methylation clusters for each year. Statistically significant differences were reported if the *p*-value was <0.05. 

### 2.5. Immunohistochemistry

The tissue samples were re-evaluated according to the WHO classification of endocrine and neuroendocrine tumors from 2017. Tissue samples were analyzed with immunohistochemical staining for transcription factors PIT1, SF1, and TPIT (primary antibody dilution of 1:500 for PIT-1 (Abcam ab272639) and SF-1 (Abcam ab217317), and 1:1000 for T-PIT (Atlas Antibodies AMAb91409)) [2,31]. Sections (3 μm) of FFPE tissue blocks were cut on a microtome and placed on glass slides. Tissue sections were deparaffinized, and the standardized protocol included heat-induced epitope retrieval (HIER) by incubation in tris-based cell conditioning 1 (CC1). After HIER endogenous peroxidase activity was blocked by incubation with 1.5% hydrogen peroxide, the sections were incubated with a primary antibody to detect the protein of interest. The antigen–antibody complex was detected by using appropriate secondary antibodies directed against the species of the primary antibody and conjugated to peroxidase complexes using OptiView (Ventana Medical Systems, Inc., Marana, AZ, USA). The staining was visualized with 3.3′ diaminobenzidine (DAB) as the chromogen. Finally, the sections were counterstained with Mayer’s hematoxylin (Ventana ref. 790-2208). 

Paraffin sections of tissue microarrays comprising normal tissue and tissue from the adrenal gland, testis (express SF1), tonsil, colon, and different subtypes of PAs expressing PIT1 and TPIT were used as positive controls. Primary antibody omission was used as the negative control. 

## 3. Results

### 3.1. WHO Classification from 2017

Of the 54 patients initially included in the study (Figure 1), 12 patients were excluded: 4 patients had incomplete postoperative MRI follow-up, 4 patients had insufficient tissue quality for DNA methylation analysis, and 4 patients had poor-quality data according to the DNA methylation quality control report.

Of the remaining 42 patients, 16 (38%) were females and 24 (62%) were males. The median age at surgery was 61 years (range of 26–87). Immunohistochemical staining for transcription factors revealed that 29 PA samples (69%) were positive for SF-1, 5 samples (11.9%) were positive for TPIT (indicating silent ACTH-secreting pituitary adenomas), and 1 sample (2.4%) was positive for PIT1. Only one adenoma (2.4%) was classified as a null-cell pituitary adenoma. In six cases, minimal tumor tissue was available, which necessitated a DNA methylation analysis to be performed without immunohistochemical staining; however, all PAs included in the study were nonfunctioning, as none of the eight TPIT- or PIT1-positive adenomas presented with clinical Cushingoid symptoms or acromegalic features. 

Invasive growth was found in 18 (42.9%) tumors based on the Hardy and Knosp classifications. The mean preoperative tumor volume was 7.9 cm^3^ (95% CI: 4.9–10.9), and the mean postoperative tumor volume was 4.4 cm^3^ (1.2–7.6).

### 3.2. DNA Methylation

From an unsupervised hierarchical cluster analysis (UHCA) of the 42 samples, the samples were clustered according to transcription factor staining. One chief cluster contained all SF-1-positive samples, while two minor clusters contained TPIT- and PIT1-positive samples (Figure 2). Sample 42 clustered adjacent to TPIT-lineage samples, but showed IHC staining for SF1; therefore, this sample was excluded for further analysis. 

Of the included samples, 33 (78.6%) were SF1-lineage NFPAs. When analyzed via a bootstrap analysis, these showed identical hierarchical clustering patterns to those prior to the exclusion of the TPIT-lineage and PIT1-lineage NFPAs (Figure 3). This bootstrap analysis revealed an AU of 86% for each of the five minor clusters. Clusters 1, 3, and 5 included five (15.6%), four (12.5%), and six (18.8%) samples, while clusters 2 and 4 included eight (24.2%) and ten (31.3%) samples. All clusters showed invasive growth and tumors with postoperative expansion (Table 1). No statistically significant differences were found in mean pre- and postoperative tumor volumes.

These robust clusters were comparable with respect to preoperative and postoperative tumor volumes and median follow-ups (Table 1). In cluster 2, six of the eight tumors (75%) were reoperations for tumors operated on prior to 2007 (*p* = 0.013). Additionally, seven of the eight tumors (88%) in cluster 2 had a medium or high regrowth rate.

The heatmap of the 5000 most variable CpGs (Figure 4) did not reveal discernable methylation patterns unique to any single cluster, but the five samples from cluster 1 were dispersed to other clusters.

The heatmap showed no observable tendencies in CpGs (Figure 4). Each cluster is summarized here, while the complete list of DMPs, genetic overlaps, and DMRs are provided in Appendix A.

Cluster 1 had 8873 DMPs (886 with a mean differentiation over 0.3), demonstrating potential gene involvement in different signaling pathways. The DMR analysis showed six different regions involved in cluster 1 in which the CREB5 gene was identified. 

Cluster 2 showed 325 DMPs and multiple signaling pathways, including for VEGF (*p* = 2.83 × 10^-5^ and Ras (*p* = 6.56 × 10^-5^) signaling. Only two DMRs were found for cluster 2, associated with genes for ITM2C and SOCS2.

Cluster 3 had 3114 DMPs (1231 with a mean differentiation over 0.3). The genetic overlaps for these sites generally regarded the ErbB signaling pathway 6 (*p* = 9.96 × 10^-6^). In the 107 DMRs found, the GALNT17 and GAS7 genes were found multiple times. 

Cluster 4 showed 674 DMPs that were associated with different developmental pathways than cluster 2. No DMRs were found.

Cluster 5 showed no clear differential methylation trends. Two DMRs were found: LINC00886 and MYO3A. 

### 3.3. Adenoma Regrowth Potential and DNA Methylation

Based on the five clusters found through an unsupervised hierarchical cluster analysis (UHCA), Kaplan–Meier survival estimates (Figure 5) were calculated to evaluate whether these clusters had different growth rates. Regarding reintervention (radiotherapy or reoperation), the hazard ratio from a Cox regression showed no statistically significant differences between the groups (*p* = 0.62) (Figure 5A), even though cluster 1 had no reinterventions and cluster 5 had 3/6 (50%) reinterventions during the follow-up period. 

All eight tumors in cluster 2 had regrowth within 30 months (Figure 5B) compared to only two (40%) tumors from cluster 1, but the findings showed no statistically significant difference (*p* = 0.62). A low regrowth rate occurred in all groups (Figure 5C), but clusters 3 and 1 had 50% and 60% of tumors show a low regrowth rate, compared to only 20% and 15% for clusters 4 and 2 (*p* = 0.94). Cluster 4 had no tumors show a high regrowth rate, while the other four clusters did have tumors with a high regrowth rate, but this was again nonsignificant (Figure 5D) (*p* = 0.68).

The mixed effects model, with which to compare the accumulated regrowth over 8 years (the median follow-up for the cohort was 75 months) for the five methylation clusters, showed differences, especially for cluster 1 (Figure 6). Statistically significant differences were found as follows: 

Between clusters 4 and 5 at one year after the postoperative scan (*p*-value = 0.03).

Between clusters 1 and 2 at two, three, and four years after the postoperative scan (*p*-values = 0.03, 0.02, and 0.03).

Cluster 2 showed significant regrowth compared to clusters 3 and 5 at four years after the postoperative scan (*p*-values = 0.02 and 0.03). 

Cluster 3 showed significant regrowth to clusters 1, 2, and 4 at seven years after the postoperative scan (*p*-values = 0.03, 0.01 and 0.03). 

Cluster 2 showed significant regrowth to clusters 1 and 3 at eight years after the postoperative scan (*p*-values = 0.04 and 0.03).

## 4. Discussion

In this study, we investigated whether DNA methylation subgroups of NFPAs correlated with tumor regrowth potential. Among SF1-lineage NFPAs, we found five methylation clusters, each with differential methylated sections that corresponded to different intracellular signaling pathways. The clusters showed no statistically significant differences in accumulated regrowth percentages, but there were tendencies for some subsets of adenomas to have a higher regrowth potential. While the statistical nonsignificance is most likely due to low sample sizes, the findings suggest a clear potential for DNA methylation profiling to serve as a prognostic marker for regrowth.

Our analysis revealed robust methylation clusters for NFPAs based on genome-wide methylation patterns. To our knowledge, these subgroups have not been described before. While it is known that NFPAs have very different regrowth potentials (probably due to intertumoral biological heterogeneity) [13,21,32,33,34], no common genetic or epigenetic differences between NFPAs have previously been correlated with quantitative tumor regrowth potential and clinical data. 

Although we initially intended to include only SF1-lineage NFPAs in this study, eight TPIT- or PIT1-lineage adenomas were also described, as their original diagnosis was based on the WHO 2004 classification for PAs. These adenomas were all clinically silent. This subgrouping for PAs based on DNA methylation is comparable to that of two previous studies [13,14] that showed clear correlation between DNA methylation patterns and immunohistochemical staining for transcription factors determining the lineage of origin for PAs. Thus, the inclusion of TPIT- and PIT1-lineage NFPAs in our study validates our data analysis approach, indicating consistency with previous findings in subgrouping PAs according to transcription-factor-specific adenohypophyseal lineage.

Hallen et al. [21] found 605 DMPs between radiologically stable and reintervention tumors. These differences are partly in line with our findings, as the most comparable groups of our cohort showed similar differentiation (clusters 2 and 3). Our cluster 1 had the most DMPs (8873), and when the mean difference in beta values was set at 0.3, cluster 1 had 886 unique DMPs. When the same cutoff was used by Hallen et al., the number of DMPs detected was reduced from 47,680 to 605. The many DMPs found for cluster 1 could explain why this cluster was dispersed among the other clusters on the heatmap, as the cutoff for this analysis was set at 5000 CpGs.

While we located none of the same genes as Silva-Junior et al. [13], Mosella et al. [14], or Hallen et al. [21], we did identify similar signaling pathways (cell junction, metabolic process, and neuronal system development). Interestingly, cluster 2 showed higher regrowth potential compared to the other clusters, and also showed links with the VEGF and Ras signaling pathways, both of which are associated with aggressive regrowth for pituitary adenomas according to a review by Serioli et al. [35], with VEGF as a potential target [36]. One DMR for cluster 1 was associated with the CREB5 gene, which has previously been associated with the regulation of cell growth and proliferation in glioma stem cells [37]. Similarly, GALNT17 and GAS7 were found for cluster 3. GALNT17 has been associated with centromeric instability, leading to brain pathology [38], and GAS7 has been reported in cells in growth arrest while functioning as an inhibitor for neuroblastoma [39], so the deletion of this gene promotes metastasis. Only two DMRs were found for cluster 2, one of which was related to the SOCS2 gene that can regulate the JAK–STAT pathway. SOCS2 gene mutation has been identified in tumor tissue [40]. A DMR associated with the LINC00886 gene was found for cluster 5. This gene has been reported to be involved in the regulation of malignancy for both thyroid cancer and laryngeal carcinoma, specifically DNA hypermethylation silencing [41,42]. 

A significant challenge lies in the evaluation of large-scale multidimensional DNA methylation datasets. Emerging studies have begun incorporating the use of artificial intelligence (AI) and multifocal testing, which is generating new insights into the complicated intracellular signaling pathways [43]. In this study, we performed our cluster analysis in two ways: firstly, based on all probes on the EPIC chip, as by Silva-Junior et al., and secondly, only on the 1000–5000 most differentiated probes, as has done by others [20]. Both approaches generated similar results regarding distribution in heatmaps and in relation to clinical outcomes.

From a patient perspective, a major concern after trans-sphenoidal PA surgery is whether a reintervention will be needed in the future. Based on DNA methylation profiling, we were able to generate five different clusters of NFPAs. These clusters showed different trends regarding the need for reintervention. While no reintervention was needed in cluster 1, some of the adenomas here had high regrowth rates and could have required reintervention after the study period. One patient in cluster 4 underwent reoperation due to regrowth, which is a competing factor as it may have reached 100% regrowth if left untreated. A previous study by Hallen et al. also described a connection between DNA methylation grouping and reoperation.

The median follow-up time in the current study was 75 months (6 years and 3 months). In Hallen et al., the follow-up period was five years, but patients with tumor growth without reintervention were excluded. This underlines the heterogeneity in PAs, as some show regrowth that is of less clinical importance. The authors did not state how many patients were excluded, however. Comparable to our study, 75% of cases showed regrowth, with only six patients needing reintervention. A follow-up longer than 75 months should have been possible for our study, but loss to follow-ups means that longer follow-up periods could result in selection bias. A newly published systematic review of patients operated on for PAs [23] showed that tumor recurrence rates increased proportional to the length of follow-ups, e.g., a 33% recurrence rate after 10 years for NFPAs. This study also reported a lower risk of developing symptoms for patients operated on for incidentalomas and microadenomas compared to those with macroadenomas never undergoing surgery. The findings of the current study suggest that longer follow-up periods are necessary to distinguish between subgroups showing slow or no regrowth, such as clusters 1 and 3. 

This study adds to the growing research and previous studies addressing the need to explore the epigenetic differences in NFPAs. It is the first study to change the narrative of tumor regrowth from a binary outcome measure to a continuous outcome measure (tumor regrowth potential). It raises the possibility of differentiating the treatment strategies for patients with NFPAs, as the follow-up regime may be determined at least in part based on the epigenetic features of each individual tumor. For example, it appears that PAs with the epigenetic features of cluster 2 might benefit from more frequent clinical follow-ups or even earlier referral to radiotherapy, as this subgroup of PAs potentially do not benefit from repeated surgery.

### Limitations

The major limitations of this study are sample size and follow-up time. Considering that 33 patients with an SF1 lineage were allocated to five subgroups, the correlation between changes in the regrowth rate and changes in methylation patterns need to be very strong for a statistically significant difference to show. This is enhanced by the relatively short follow-up time of 75 months where only a few events (reintervention, regrowth) occurred. 

Furthermore, the current analytical methods for evaluating DNA methylation data are not optimal. Our findings may have been different if other analytical packages had been used to analyze the DNA methylation patterns. Previous studies have explored various analytical approaches [13,14] to reach a consensus of how to analyze data and present them in the most objective, software-independent fashion.

## 5. Conclusions

DNA methylation profiling appears to have potential in the prognostication of regrowth potential in NFPAs. Our results need further validation in larger study populations and with longer follow-up times to further explore the regrowth potential within the heterogeneous group of NFPAs. This will hopefully enable clinicians in the future to identify the patients that need a more extensive follow-up from patients with a low risk of adenoma recurrence.

## Figures and Tables

**Figure 1 cancers-16-02210-f001:**
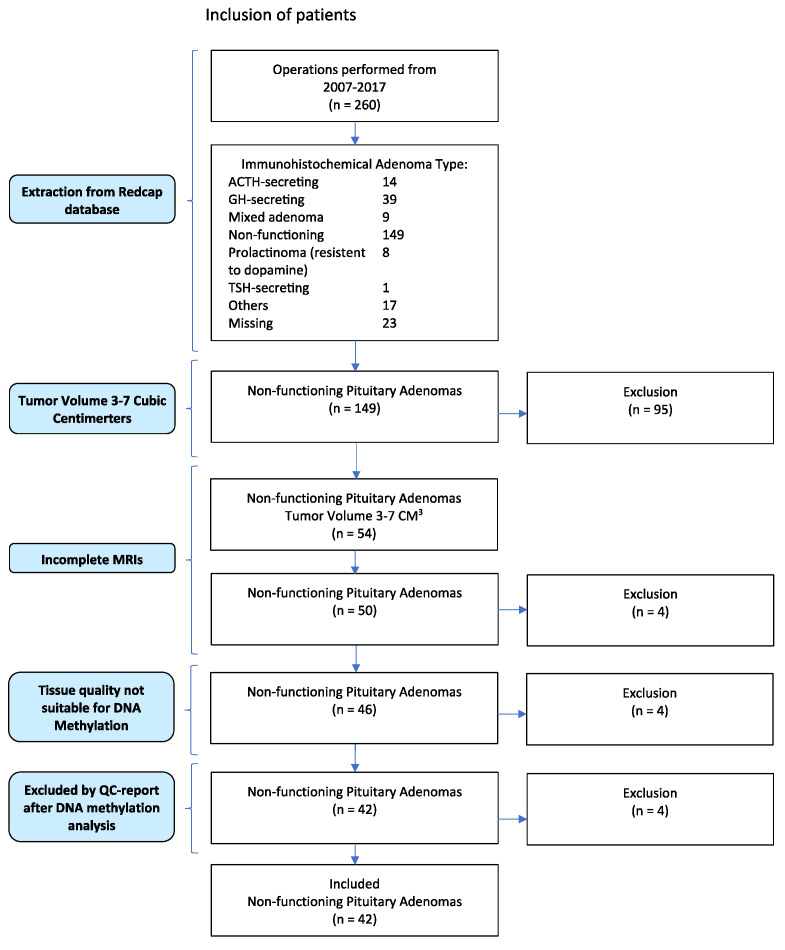
PRISMA flowchart of patient inclusion.

**Figure 2 cancers-16-02210-f002:**
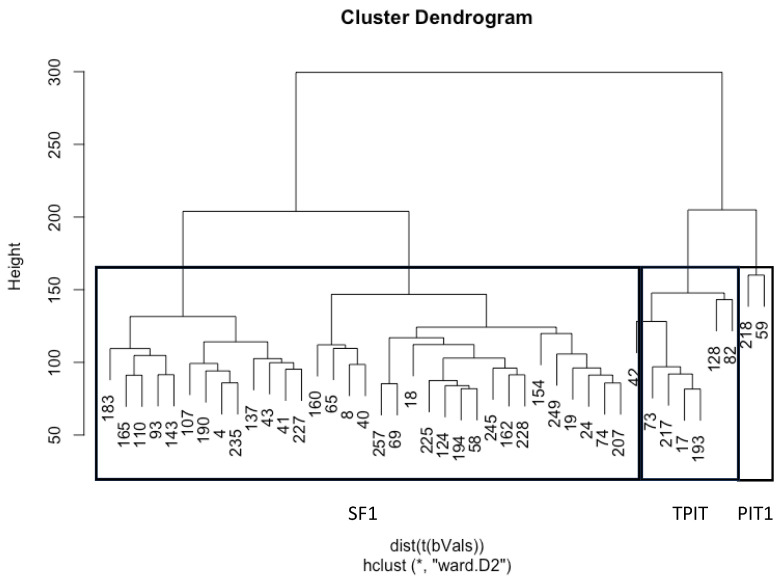
Unsupervised hierarchical cluster analysis (UHCA) of 42 tissue samples using full-genome (* is a symbol for all 766,404 CpGs included) methylation profiles. This shows distinct clustering that correlated with immunohistochemical staining for transcription factors SF1, TPIT, and PIT1.

**Figure 3 cancers-16-02210-f003:**
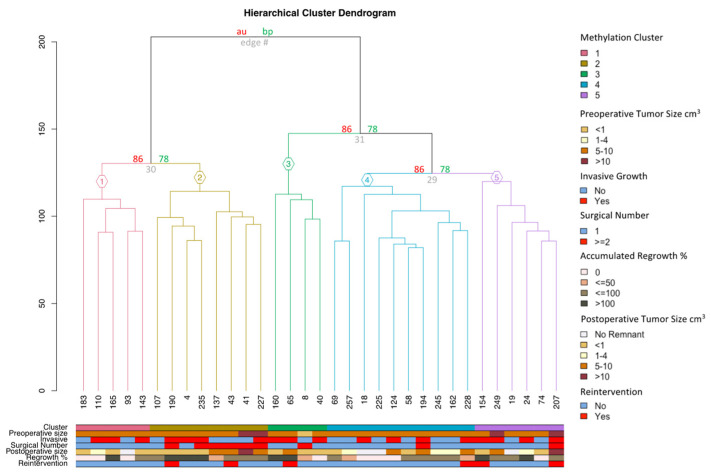
Unsupervised hierarchical cluster analysis (UHCA) via a x200 bootstrap resampling procedure on 33 SF1-lineage NFPAs. Results demonstrate five robust clusters (AU = 86%), defined as 1–5. Pre- and postoperative tumor sizes (cm^3^) were grouped, while invasive growth, primary or repeated surgery, and the need for reintervention used binary outcomes (yes/no). Accumulated regrowth percentage was grouped into categorical values.

**Figure 4 cancers-16-02210-f004:**
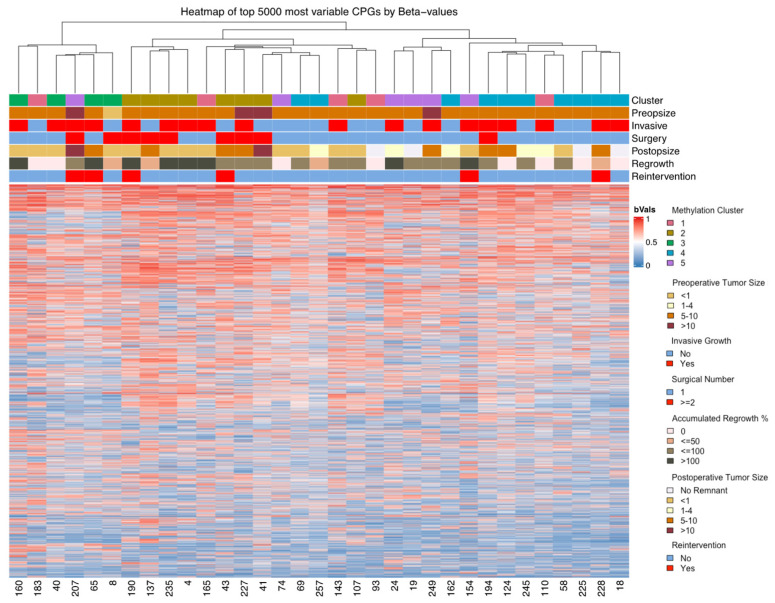
A heatmap illustrating the top 5000 most variable CpGs based on beta values reveals clustering patterns that align with both the comprehensive genomic analysis and clinical data. Notably, cluster 1 exhibits a notable reorganization relative to other clusters, stemming from its greater variability in differential methylation positions (DMPs) compared to the remaining clusters; however, discernible methylation patterns among the other clusters are not readily apparent.

**Figure 5 cancers-16-02210-f005:**
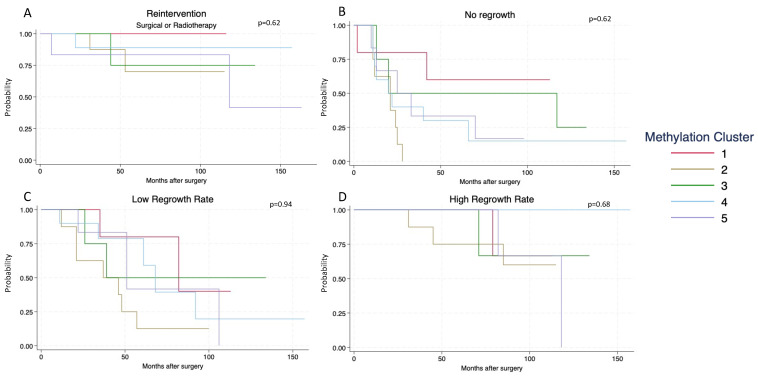
Survival analysis based on Kaplan–Meier estimates: (**A**) for patients undergoing reintervention (reoperation or radiotherapy) during the follow-up period; (**B**) for tumor remnants showing no increase in volume; (**C**) for tumors with more than a 50% increase in volume; and (**D**) for tumors with more than a 100% increase in volume during the study period.

**Figure 6 cancers-16-02210-f006:**
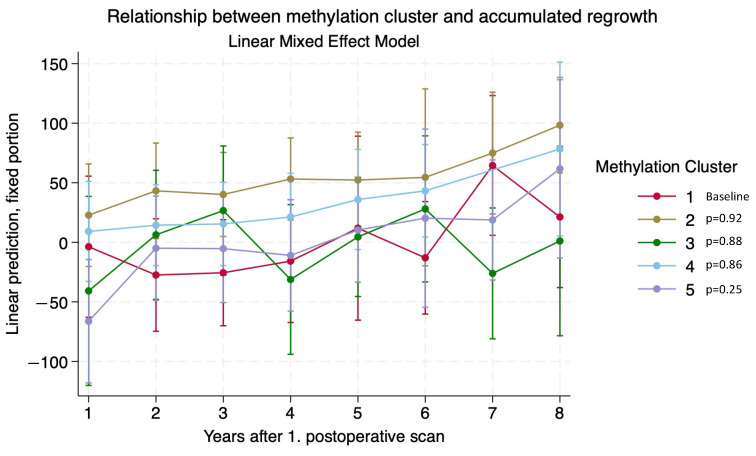
Linear mixed effects model of methylation profiles and regrowth potential, showing clear tendencies for different regrowth patterns dependent on methylation profiles. Cluster 2 has apparent, but statistically insignificant, regrowth potential compared to clusters 1 and 3.

**Table 1 cancers-16-02210-t001:** Characteristics of 42 nonfunctioning pituitary adenomas, including 33 of an SF1 lineage that are grouped according to methylation cluster.

n		Methylation Clusters of SF1-Lineage NFPAs					*p*-Value
	All	1	2	3	4	5	
Total (%)	42 (100)	5 (15.6)	8 (24.2)	4 (12.5)	10 (31.3)	6 (18.8)	
Age (median) (min–max)	61 (26–87)	63 (43–77)	59 (53–68)	62.5 (50–80)	53 (26–87)	66 (49–76)	*p* = 0.83
Invasive growth	18 (42.9)	3	4	3	4	4	
Regrowth Rate No Low Medium High	NA	3 0 1 1	0 1 4 3	1 1 1 1	2 3 5 0	1 1 2 2	
Radiotherapy	2 (4.8)	0	1	0	0	1	
Re-operations	6 (14.3)	0	2	1	1	2	
Mean preoperative tumor volume (cm^3^) (95%CI)	7.9 (4.9–10.9)	6.06 (5.6–6.5)	6.2 (4.6–7.9)	5.9 (2.9–8.8)	6.1 (5.1–7.1)	9.1 (4.8–13.2)	*p* = 0.38
Mean postoperative tumor volume (cm^3^) (95%CI)	4.4 (1.2–7.6)	1.2 (0.4–2.1)	3.5 (1.6–5.3)	3.6 (1.1–6.0)	1.9 (0.8–3.1)	4.8 (–0.3–10.0)	*p* = 0.26
Median follow-up time (months) (min–max)	75 (12–167)	82 (66–130)	88 (45–159)	96 (58–147)	60 (12–157)	86.5 (46–167)	*p* = 0.39

## Data Availability

All data are available upon request to the corresponding author.

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
