# Peer review of "Genome-Wide DNA Methylation Profiling as a Prognostic Marker in Pituitary Adenomas—A Pilot Study"

_cancers, 2024, doi:10.3390/cancers16122210_

Round 1
Reviewer 1 Report
Comments and Suggestions for Authors
Møller and colleagues analyzed a cohort of non-functioning pituitary adenomas (NFPA) to study the global DNA methylation, using the EPIC methylation array. They confirmed the already-known clusters of tumor samples associated with histology subtypes/immunohistochemical staining. The novelty of this study was the analysis of a particular subtype (SF1-lineage) of NFPA to define the association between methylation profiles and the regrowth risk. The results suggested subsets of tumor samples with higher regrowth potential and indicated possible markers.
I have some comments:
- Since postoperative residual tumor tissue increases the risk of regrowth, how does postoperative size impact the difference observed specifically for cluster 2 for which 50% of samples had a post-operative size higher than 5 cm3? The authors stated that no statistically significant differences were found. Please explain and show the analysis done.
- Please clarify how Differentially Methylated Probes (DMPs) were defined: are there control samples used in this study? Or were DMPs defined based on the comparison among clusters?
- In figure 2, do the squares indicate the IHC subtypes? Which was the IHC staining of sample 42?
- In figure 3, please make the bottom bars bigger and order the legend on the right as done for the bottom one.
- A typo was on page 2, line 72: “promoters”; another on page 6, line 234 “Table 1”.
Reviewer 2 Report
Comments and Suggestions for Authors
1. Have you uploaded your raw data on NCBI? Kindly write the accession no in the manuscript.
2. What is the significance of DNA Methylation and gene expression in your Data?
3. What methodology you have adopted while calculating the sample size in this manuscript?
4. You have done an enrichment analysis but we did not find any results regarding this in the main manuscript.
5. What is the relevance of doing Immunohistochemistry you haven't described any results from this experiment in this manuscript.
Round 2
Reviewer 2 Report
Comments and Suggestions for Authors
All the comments were incorporated in this manuscript.Now it seems fine.